# Resistome metagenomics from plate to farm: The resistome and microbial composition during food waste feeding and composting on a Vermont poultry farm

Korin Eckstrom[1], John W. Barlow[2]*

**1** Department of Microbiology and Molecular Genetics, The University of Vermont, Burlington, Vermont, United States of America, **2** Department of Animal and Veterinary Sciences, The University of Vermont, Burlington, Vermont, United States of America

* john.barlow@uvm.edu

**Data Availability Statement:** All data files are available in the Sequence Read Archive (SRA) or BioProject under the accession number PRJNA549056.

## Abstract

Food waste diversion and composting, either mandated or voluntary, are growing alternatives to traditional waste disposal. An acceptable source of agricultural feed and composting material, methane-emitting food residuals, including post-consumer food scraps, are diverted from landfills allowing recapture of nutrients that would otherwise be lost. However, risk associated with the transfer of antimicrobial resistant bacteria (ARB), antibiotic resistance genes (ARGs), or pathogens from food waste is not well characterized. Using shotgun metagenomic sequencing, ARGs, microbial content, and associated virulence factors were successfully identified across samples from an integrated poultry farm that feeds post-consumer food waste. A total of 495 distinct bacterial species or sub-species, 50 ARGs, and 54 virulence gene sequences were found. ARG sequences related to aminoglycoside, tetracycline, and macrolide resistance were most prominent, while most virulence gene sequences were related to transposon or integron activity. Microbiome content was distinct between on-farm samples and off-farm food waste collection sites, with a reduction in pathogens throughout the composting process. While most samples contained some level of resistance, only 3 resistance gene sequences occurred in both on- and off-farm samples and no multidrug resistance (MDR) gene sequences persisted once on the farm. The risk of incorporating novel or multi-drug resistance from human sources appears to be minimal and the practice of utilizing post-consumer food scraps as feed for poultry and composting material may not present a significant risk for human or animal health. Pearson correlation and co-inertia analysis identified a significant interaction between resistance and virulence genes ($P = 0.05$, $RV = 0.67$), indicating that ability to undergo gene transfer may be a better marker for ARG risk than presence of specific bacterial species. This work expands the knowledge of ARG fate during food scrap animal feeding and composting and provides a methodology for reproducible analysis.

**Funding:** KE and JWB received funding from the United States Department of Agriculture Northeast Sustainable Agriculture Research & Education (USDA-NE-SARE; https://www.northeastsare.org/) Graduate Student grant (project #GNE16-117). The funders had no role in the study design, data collection and interpretation, or the decision to submit the work for publication.

**Competing interests:** The authors have declared that no competing interests exist.

# Introduction

The global crisis of antimicrobial resistance (AMR) has been attributed to antimicrobial overuse, improper prescribing, extensive use as growth promoters in agriculture, and the slowing development of new antimicrobials [1]. As we continue in the "post-antibiotic era", increasing pressure is placed on proper stewardship and surveillance efforts. In particular, environmental and agricultural reservoirs of resistance have been identified as key points of intervention. However, this work has focused primarily on soils, wastewater, and manures. An additional component of the agricultural cycle is the fate of food wastes and residuals, however investigation into the contribution of these materials to AMR persistence or introduction is limited.

Diversion of food scraps to agriculture is not only a sustainable practice, but in states such as Vermont it is being promoted as an alternative to meet current regulations. In the wake of Vermont's Universal Recycling Law (Act 148) [2] and similar legislation in other states or municipalities, the fate of microbial species in food waste and residuals is under scrutiny; agricultural composts and soils represent a major contact point between the environment, animals, and humans, yet the extent of novel bacteria and associated antimicrobial resistance genes (ARGs) in comingled food residuals is unknown. Poultry farms may represent increased risk, as in contrast to the regulation of feeding of food waste to species such as swine, in Vermont there are no prohibitions on using raw food scraps as poultry feed[3].

As legislation implementing food waste composting and diversion becomes more popular, risk assessment of food wastes and residuals must be performed. Mandates such as Vermont's Universal Recycling Law (Act 148) suggests these materials might be used for agricultural feed and composting, particularly within the poultry production chain, but also for energy production on farms that utilize anaerobic digesters. Previous work has shown that both AMR microorganisms and ARGs exist in food products [4–7] at the point of consumer purchase or within households, which are also the largest producers of food wastes [8]. These comingled food residuals may carry antimicrobial resistant bacteria and genes from multiple sources, yet their fate once they are incorporated into the farm setting is unknown.

Current guidelines for feeding food wastes to commercial poultry operations recognize the risk of pathogen introduction. Few restrictions exist for feeding food waste to chickens, and to our knowledge, no regulations address the potential transmission of ARGs from food waste to livestock. As there is direct contact between the "vehicle" (food waste) and the animal, a potential new source of antimicrobial resistance in the food cycle is born from implementing post-consumer food scrap feeding on commercial poultry farms.

Assessment of ARB and ARGs has been performed in similar materials, such as swine or dairy cattle manures [9–14], yet the extent and relative importance of food scraps as a source of resistance is largely unknown. The purpose of this study to identify the range and magnitude of ARGs in food scraps received by an integrated poultry farm and composting operation. Samples of post-consumer food wastes and residuals were collected at the sources and across the farm system, from importation and use as poultry feed, to the finished composts and egg products.

Traditional approaches to resistance monitoring or risk assessment have utilized culture-based techniques or lower-throughput culture-independent strategies such as qPCR. In this study, we utilized shotgun metagenomic sequencing to assess both the bacterial and resistance gene diversity throughout the food scrap acquisition, feeding and post-feeding composting processes. This technique has recently been used to investigate the resistome of sources such as manures, agricultural soils, lakes, and hospital effluents [11,15–17]. Additionally, the use of cloud-based bioinformatics resources showcases the accessibility of these tools for ARG surveillance for projects of any scale.

In this study, the focus is placed on the potential impacts of human food waste composting on the poultry farm resistome, as well as commercial products leaving the farm. The primary aim of this work was the identification and characterization of antibiotic resistance genes (ARGs) in food wastes, composts, and farm products, which was achieved using shotgun metagenomics methods. Additional aims include the assessment of microbial communities and potentially pathogenic species, as well as associated virulence factors from all samples to elucidate the potential mechanisms of resistance transfer within the farm environment. Finally, the relationships between these genes and bacterial communities were successfully investigated and the results provide insight to potential avenues for future studies of food waste management practices.

## Materials and methods

### Collaborating farm

All data were obtained from a single commercial diversified farm located in northeastern Vermont (44˚32'56.9"N 72˚13'52.5"W). The study farm produces eggs for wholesale and direct to consumer retail sales, seasonal produce, and soil amendments including compost and vermiculture worm castings. With the exception of feeding post-consumer food waste, which can not be certified as an organic feed, the farm follows USDA National Organic Program standards for husbandry practices, including no use of antimicrobials. The farm provides a commercial compost collection service collecting food scraps and food manufacturing wastes from regional businesses and institutions.

### Sample collection

Samples were collected both on-farm and at individual food scrap collection sites in February 2017. On-farm samples included i) raw food scraps (RFSC); ii) three stages of windrow composting piles: raw compost (RWCO), unfinished compost (UFCO), and finished compost (FICO); iii) three stages of worm casting: the initial layer of substrate (TWCA), immediately after sifting (SWCA), and the packaged commercial product (WOCA); and iv) eggs from the laying hens within the barn, including outer wash as a representative of the barn environment (EGWA) and shells to represent composition upon leaving the farm (EGSH). Off-farm samples were taken as representatives from each bin present at the site, including a regional school district kitchen (SCHO), outpatient hospital kitchen (HOSP), nursing home kitchen (NURH), and grocery store (GROC). A visual representation of the sampling scheme is shown in Fig 1. Additionally, a blank sample (TRBL) was included in all analysis to capture any noise generated from environmental or reagent contamination. For each substrate type, four sterile RNA/DNA free 50 mL conical tubes (Ambion, Thermo Fisher, Waltham, MA) were filled using grab sampling across various depths and locations of on-farm piles or across bins at collection sites. However, due to the time of year, much of the substrate was frozen and this did impact the ability to sample more than a few inches into the core of outdoor samples. For eggs, three eggs were taken directly from hen houses within the barn. All samples were transported on ice back to the University of Vermont and stored at -80˚C until further processing and DNA extraction.

All samples were collected in compliance with University of Vermont Research Protections Guidelines. Because the study involved only collection of food waste, compost materials and eggs, no human subjects, vertebrate animal, or biosafety oversight protocol approvals were required.

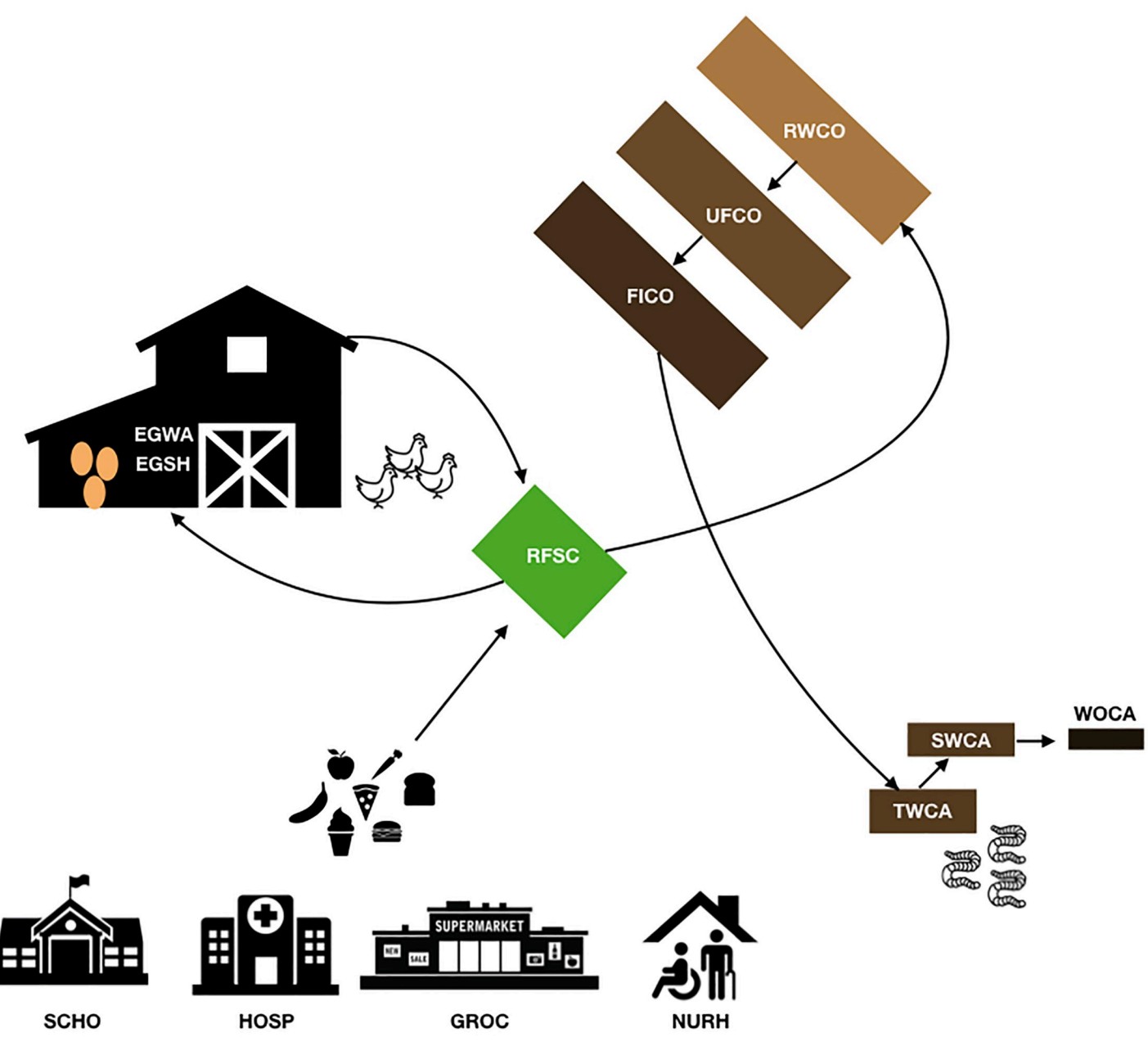

**Fig 1. Map of sampling scheme and directionality of food waste movement on the farm.**

### Pre-processing and DNA extraction

Due to the nature of food scrap samples, significant efforts were put into the "pre-processing" of all samples to reduce the amount of eukaryotic DNA contamination. Eukaryotic genomes, particularly those of plants, are orders of magnitude larger than microbial genomes and present a challenge to profiling intermingled food wastes. Physically removing these cells prior to DNA extraction (described here as "pre-extraction processing") allows for shallower sequencing and a reduction in the need to bioinformatically remove potential host sequences (human, chicken, crops, etc.) saving on costs at each step. To accomplish this, we developed physical host tissue depletion and DNA enrichment and extraction methods in a series of preliminary

experiments comparing variations in agitation and filtration using a series of compost samples collected in the months prior to formal sample collection in February. Optimization of DNA yield and host cell removal following tissue pre-extraction processing was determined by comparing crude estimates of genomic DNA and 16S rRNA gene (primers 27-F and 519-R, [18]) to 18S rRNA gene (primers EK1-F and EK1250-R [19]) amplicon yields on DNA extracted from compost samples using 3 different filter sizes (20, 40, or 60 micron) and 3 different commercially available DNA extraction kits. The combination of a 40 micron filter and DNA extraction using the Qiagen (formerly MoBio) PowerSoil kit (Qiagen, Hilden, Germany) resulted in the consistently highest crude genomic DNA yields.

These pre-extraction processing procedures (e.g. physical agitation and vacuum filtration) were then completed for the study samples. Briefly, 1 g of each sample was added to 10 mL sterile UltraPure water (Thermo Fisher, Waltham, MA) in a 50 mL conical Tube (Ambion, Thermo Fisher, Waltham, MA). A total of four tubes were prepared for each sample. Sterile water was warmed to 42˚ C to improve bacterial disruption upon vortexing. This was performed for all samples except the egg shell and egg wash. For egg wash, whole eggs were placed into individual sterile Whirl-Paks with 40 mL of sterile, warmed water and gently shaken for 2 minutes. Wash material was then placed into a sterile 50 mL conical for further processing. Once washed, for egg shell samples, the eggs were cracked on the edge of a sterile beaker and all interior products were discarded. Any remaining albumin was rinsed thoroughly with additional sterile water. The shell was then crushed with a gloved hand and inserted into a sterile 50 mL conical tube with 40 mL of warmed (42˚ C) sterile water and agitated/crushed for 2 minutes with a sterile glass rod (method adapted from a previous study [20]).Once prepared, all sample mixtures were transferred to a multitube vortexer and shaken for 5 minutes at 1500 rpm to disrupt bacterial adhesion to any food scraps or soil particles. All samples were then filtered through a 40 micron SteriFlip (Millipore Sigma, Darmstadt Germany) tube using vacuum filtration and combined into a single 40 mL volume per sample type. This was then centrifuged for 15 minutes at 2,000 g to pellet biological material. Supernatant was discarded and pellets were resuspended in 800 μL of sterile water prior to DNA extraction. Samples were stored at -20˚ C if not immediately used for extraction.

DNA was extracted from sample filtrates using the Qiagen PowerSoil kit. Manufacturer's protocol was followed with the following changes. Briefly, instead of unprocessed soil, for each sample 400 μL of the pre-processed liquid filtrates was added to a sterile tube containing beads. Total DNA was eluted and stored at -20˚ C until quantitation and sequencing.

The concentration of DNA in each sample was quantitated using the Qubit 2.0 dsDNA BR Assay system (Thermo Fisher, Waltham, MA). The manufacturer's protocol was followed and 1 μL of sample DNA to 199 μL of working solution was used. Concentrations ranged from <0.025 ng/μL in the trip blank to 13.5 ng/μL in the finished compost, with an average of 3.7 ng/μL in experimental samples.

## Library preparation and shotgun metagenomic sequencing

Library preparation and sequencing was performed at the UVM Cancer Center Advanced Genomics Lab (Burlington, VT). DNA quality was assessed, and fragmentation was performed using the Bioanalyzer system and Covaris respectively. A total of 2 ng of DNA from each sample was used for library preparation using the Nextera reagent kit (Illumina Inc., USA). All libraries were checked for quality using the Bioanalyzer system prior to sequencing. All 14 samples (13 samples + 1 trip blank) were sequenced via 100 bp single end (SE) Illumina HiSeq shotgun sequencing. Two lanes in total were used, from different flow cells and on different days, as technical replicates as well as to increase the total sequencing depth.

Initial sequence analysis was performed by the UVM Bioinformatics Shared Resources (Burlington, VT). This included demultiplexing (assigning reads to their sample using the barcodes from the library preparation stage), quality checking using FastQC [21] and storage on a remote server (Vermont Advanced Computing Core, VACC). Once sequences were retrieved, quality was examined using FastQC output files. Average sequence quality was above Q 30 for all samples, indicating that both lanes had high-quality sequences.

## Sequence analysis

The CosmosID (CosmosID Rockville, MD) software suite was used for both identification and classification of functional genes and bacterial content in all samples. Briefly, CosmosID is a cloud-based platform that uses curated reference datasets to rapidly assign metagenomic reads to the species, sub-species, and even strain level, as well as a wide array of virulence factors, antimicrobial resistance genes, and other functional databases. This is accomplished using two main algorithms, the first of which is the 'pre-computation phase that constructs a whole genome phylogeny tree using sets of fixed length n-mers (referred to as biomarkers) from the curated database. Once constructed, the second 'per-sample phase' searches metagenomic reads from submitted samples against the biomarker 'fingerprints' for identification. Resulting statistics are aggregated to maintain overall precision and allow for sample composition, including relative abundance estimates, frequency of a biomarker hit, total coverage of the reference sequence (Total Match %), and total coverage of unique biomarkers (Unique Match %). For this study, frequency and total reads were used to calculate further metrics for analysis.

Results of alignment to CosmosID databases Bacteria Q3 2017, Antibiotic Resistance Q4 2016, and Virulence Factors Q4 2016 were exported in .csv format for additional analysis in R (version 3.4.3). Previous studies utilizing shotgun metagenomics have noted that reads associated with reagent contamination can occur [22,23] and contributes to potential false positives within shotgun sequencing datasets. As a result, filtering was conducted by using all results from the trip/extraction blank (TRBL). Briefly, any samples with an extract match (i.e. same strain or gene) or match on the same branch (i.e. matched to same node within the database) to those within either TRBL sample were removed from further analysis. This strategy was used as some results may simply be rare, and occurrence in a blank rather than a read threshold allows these rare results to be conserved. Additionally, redundant results in the form of repetitive branch hits may result from short or erroneous reads. For example, if a sample contained both a branch result for *Staphylococcus* and a more specific result of *Staphylococcus aureus*, the less specific branch results were removed so as to not artificially inflate sample diversity. These types of removals are responsible for the majority of filtered hits (Table 1).

Finally, an additional parameter was calculated to aid comparison of experimental samples. As each sample contained differential proportions of reads associated with Eukaryotic DNA, an abundance ratio similar to gene copy/16S rRNA copy was created. The metric allowed for a better representation of the abundance of resistance genes and virulence factors by accounting for the putative bacterial load of the sample. Abundance ratios were calculated as total bacterial hits/total reads per sample and hits/ total bacterial hits and expressed as counts/bacteria in results.

## Statistical analysis

Analysis on filtered results were performed using R (version 3.4.3), including total genes per sample, abundance ratios, and aggregation of results by sample. Heat maps of virulence factors and ARGs were generated using the pheatmap package [24] (v.1.0.12) and were scaled by row to normalize results by gene across samples. Calculations of sample diversity (richness,

**Table 1. Raw reads, unfiltered reads, and filtered hits for each sample.**

| Sample ID | Raw Reads | Reads with Bacteria Hit[a] | Total[b] Bacteria | Filtered Bacteria | Reads with ARG Hit | Total ARGs | Filtered ARGs | Reads with VF Hit | Total VF | Filtered VF |
|---|---|---|---|---|---|---|---|---|---|---|
| EGSH_1 | 4,848,612 | 555,216 | 85 | 42 | 888 | 15 | 11 | 859 | 21 | 13 |
| EGSH_2 | 3,753,273 | 428,965 | 67 | 33 | 766 | 15 | 12 | 675 | 20 | 12 |
| EGWA_1 | 13,209,748 | 1,395,355 | 138 | 67 | 2,421 | 26 | 21 | 3,543 | 47 | 29 |
| EGWA_2 | 10,830,163 | 1,155,081 | 113 | 53 | 1,962 | 21 | 20 | 3,003 | 42 | 24 |
| FICO_1 | 13,829,897 | 1,690,557 | 163 | 54 | 1,645 | 13 | 10 | 1,479 | 17 | 13 |
| FICO_2 | 11,117,824 | 1,378,774 | 121 | 54 | 1,396 | 12 | 10 | 1,260 | 17 | 13 |
| GROC_1 | 19,628,880 | 598,070 | 107 | 51 | 463 | 4 | 3 | 1,405 | 11 | 7 |
| GROC_2 | 14,748,726 | 465,153 | 73 | 33 | 362 | 3 | 3 | 1,056 | 9 | 5 |
| HOSP_1 | 10,568,228 | 3,747,477 | 119 | 72 | 1,524 | 0 | 0 | 4,944 | 11 | 6 |
| HOSP_2 | 8,466,316 | 3,071,119 | 75 | 37 | 1,331 | 0 | 0 | 4,057 | 10 | 6 |
| NURH_1 | 33,835,024 | 654,859 | 58 | 22 | 1,196 | 15 | 15 | 1,425 | 8 | 5 |
| NURH_2 | 25,097,582 | 487,950 | 48 | 17 | 957 | 14 | 14 | 1,017 | 7 | 5 |
| RFSC_1 | 9,520,651 | 692,419 | 152 | 66 | 890 | 12 | 9 | 795 | 12 | 6 |
| RFSC_2 | 7,839,471 | 575,086 | 121 | 52 | 749 | 11 | 9 | 614 | 13 | 7 |
| RWCO_1 | 6,536,095 | 422,996 | 184 | 69 | 661 | 8 | 4 | 1,231 | 12 | 10 |
| RWCO_2 | 5,918,271 | 379,633 | 163 | 62 | 606 | 7 | 4 | 1,211 | 8 | 6 |
| SCHO_1 | 5,761,907 | 666,454 | 158 | 96 | 1,055 | 9 | 7 | 3,246 | 9 | 3 |
| SCHO_2 | 6,049,062 | 764,125 | 139 | 74 | 1,192 | 8 | 8 | 3,388 | 11 | 3 |
| SWCA_1 | 7,582,754 | 157,170 | 94 | 19 | 191 | 0 | 0 | 348 | 4 | 3 |
| SWCA_2 | 6,475,126 | 132,104 | 67 | 16 | 161 | 0 | 0 | 326 | 3 | 2 |
| TWCA_1 | 7,682,146 | 312,351 | 121 | 43 | 798 | 8 | 6 | 789 | 12 | 9 |
| TWCA_2 | 6,981,512 | 283,197 | 136 | 43 | 736 | 10 | 6 | 709 | 16 | 9 |
| UFCO_1 | 13,092,602 | 1,621,983 | 318 | 151 | 1,657 | 16 | 13 | 1,515 | 23 | 18 |
| UFCO_2 | 10,375,658 | 1,300,381 | 229 | 108 | 1,283 | 12 | 9 | 1,145 | 19 | 15 |
| WOCA_1 | 9,748,285 | 297,759 | 137 | 40 | 336 | 1 | 0 | 538 | 3 | 2 |
| WOCA_2 | 7,951,244 | 243,048 | 111 | 33 | 254 | 0 | 0 | 433 | 3 | 2 |
| TRBL_1 | 1,071,666 | 209,073 | 31 | - | 10,726 | 61 | - | 1,901 | 4 | - |
| TRBL_2 | 838,122 | 156,928 | 24 | - | 8,194 | 58 | - | 1,422 | 4 | - |
| Average | 10,119,959 | 851,546 | 120 | 54 | 1,586 | 13 | 7 | 1,583 | 13 | 9 |

a. Hits refers to the total number of reads associated with each category

b. total columns indicate the total number of unique matches, i.e. total unique bacteria or genes.

Shannon, and Simpson) were performed using the vegan package [25](v.2.4–6). The metaMDS function using Bray-Curtis distances were used for NMDS ordination of bacterial communities in the vegan package.

Relationships between functional genes and bacteria were assessed by Pearson's correlation coefficient and co-inertia analysis. Correlation tests were performed using the Hmisc package [26] (v.4.1–1). Co-inertia analysis was performed using the made4 package [27]. Visualizations and figures were made using ggplot2 [28].

## Results and discussion

### Sequencing and additional read filtering

Total data generated, read numbers, and results of filtering are shown in Table 1. Total reads ranged from 3,753,273 to 33,835,024 per sample, excluding blanks. Total depth and read number did not appear to significantly impact results between samples however, as total read

number is not directly associated with bacterial reads (e.g. NURH on lane 1 versus EGWA or HOSP samples, which had vastly different total reads yet similar bacterial reads). Blank samples had lower total reads and reads associated with bacteria. After filtering, an average of 54 bacterial species, 7 resistance genes, and 9 virulence factors per sample were identified (Table 1).

Prior studies used ARDB and Resqu databases for ARG annotation for analysis [16,29]. However, source material from these studies were lake sediment and sterile swabs of paper money, respectively, which likely contain less diverse Eukaryotic DNA contamination compared to food waste and compost samples; for example, when a single eukaryotic host can be identified (i.e. human) those sequences can be filtered and removed, but this is an intensive process when dealing with an unknown number of plant genomes in composted materials. The steps taken to physically deplete these eukaryotic sequences via filtration in this study present a potential alternative to more expensive lysis buffers, probe depletion, or other chemical mechanisms of host depletion. Additionally, bioinformatic reduction of host contamination is difficult for environmental samples such as these, as the number and type of host sequences is unknown. Given the potential future use of shotgun metagenomics for large-scale surveillance efforts, efforts to reduce both reagent and sequencing costs makes these methods more attainable for many research groups and/or governing bodies. Our preliminary experiments with physical filtration were limited in scope and continued research validating physical depletion or enrichment methods is warranted.

In order to accurately and efficiently identify both ARGs and bacteria present, we used the CosmosID bioinformatic pipeline. By utilizing an algorithm based on data mining and phylogenetic approaches, rather than sequence assembly and alignment, these results were less susceptible to errors that eukaryotic sequences may have introduced during contig or genome assembly. This approach allows for better coverage of individual genes given the relatively short sequences generated by shotgun sequencing. CosmosID databases are heavily curated and updated, including over 150,000 bacterial genomes, and recently ranked highest in sensitivity and accuracy when compared to other popular metagenomic analysis tools [30]. Additionally, CosmosID has been applied to the analysis of environmental reservoirs such as rivers [31,32] demonstrating the breadth of reference genomes outside of purely clinical isolates.

## Characterization and persistence of ARGs

A total of 50 unique ARGs were found, ranging from 0 to 21 per sample, with individual gene abundance ratios ranging from 0 to 0.102 counts/bacteria. Total abundance ratios per sample, a proxy for overall "load" of ARGs, ranged from 0 to 0.431. Genes spanning 8 drug types were found, as well as ARGs regulating resistance mechanisms (Fig 2). Egg wash (EGWA), egg shells (EGSH), and unfinished composts (UFCO) had the most resistance genes of the on-farm samples, while the nursing home kitchen waste carried the most resistance genes of the site samples. Samples from hospital kitchen (HOSP), sifted worm castings (SWCA), and commercial worm castings (WOCA) did not have any resistance gene sequences identified after filtering.

ARG sequences related to aminoglycoside (12), tetracycline (12), and macrolide (9) resistance were found most frequently. Additionally, 10 gene sequences related to multidrug resistance (MDR) were isolated in NURH samples. Resistance genes appearing in multiple samples or of particular risk to human infection are shown in Table 2. Of these, streptomycin resistance gene *aph(6) Id* was present in the most samples, and has been previously found in wastewater [33] and lakes [16]. Several ARGs known to reside on plasmids and mobile genetic elements were found as well, including *tetM*, *tetO*, and *tetW* [34,35].

Abundance ratios of all ARGs found by sample are shown visually in Fig 3. In addition to variation in overall load, ARGs appear to cluster by sample similarity or stage of composting.

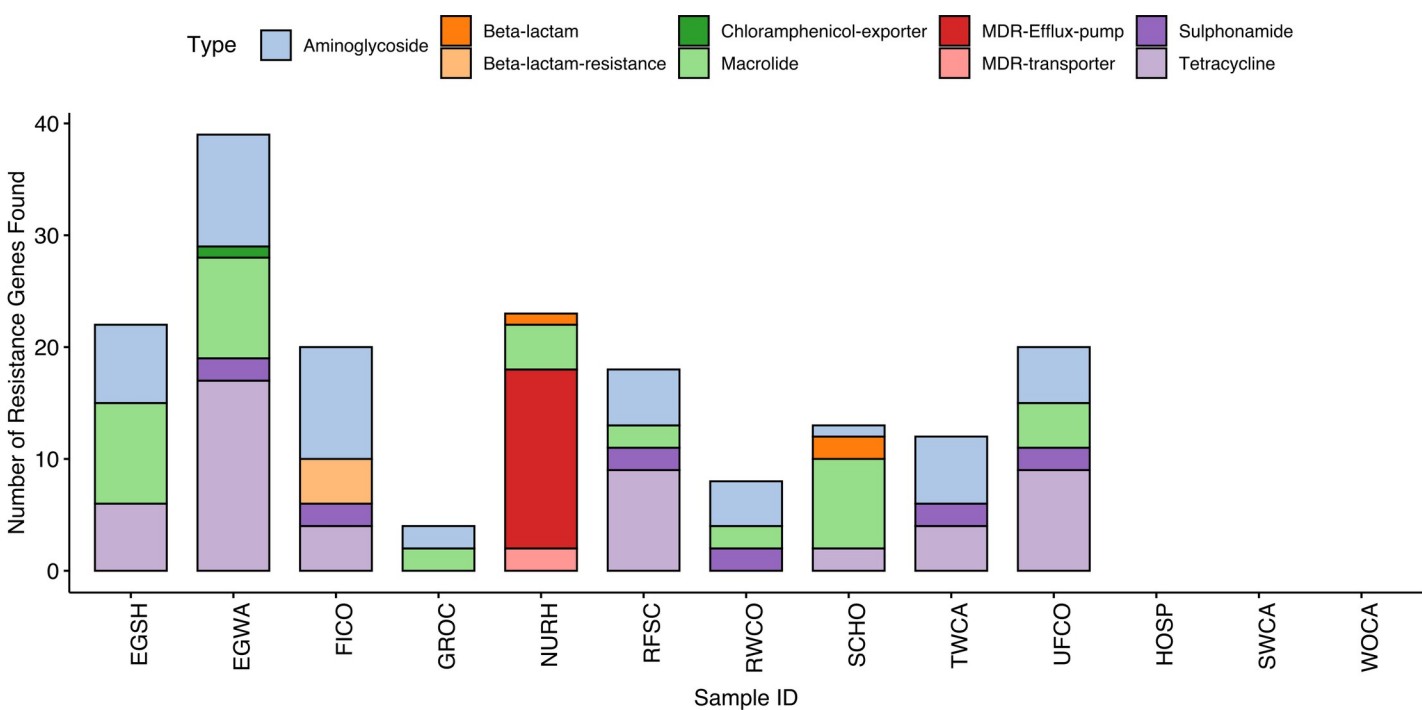

**Fig 2. Bar chart of the total number of antibiotic resistance genes (ARGs) found by drug class and sample.** In this instance, results for each duplicate were combined into a single bar.

For example, clusters are composed of samples directly related to each other, such as FICO and TWCA or RWCO, UFCO, and food scrap collection sources. This pattern is observed with the presence of specific genes themselves. Tetracycline resistance genes *tetH/L/M/O/W/X* are all present in both the raw food scraps and egg samples, while genes such as *lmrD* are only present in off-farm food waste collection sites. Macrolide resistance genes, such as *mefA/mel*, *msrD*, and *lmrD*, are only in egg and site samples. A similar resistome profile was detected in fecal and cecal samples from broiler chickens on nine commercial farms using the commercial feeds included different antimicrobial agents [36]. In contrast to this previous study, the farm in our study had no history of antimicrobial use in the laying hens suggesting at least some ARGs in this flock may be introduced with the food scraps. A limitation of our current study is the restricted scope of sampling across additional time points to characterize variation in ARG families in food scraps and farm sources and across different sources including the chickens, other domestic animal species and humans.

Other genes appear to be mitigated by the composting process. Tetracycline resistance genes, some of the most widespread of ARGs identified in this study, become undetected in later stages. For example, *tetH*, *tetW*, and *tetX* are all present at the raw compost stage, with *tetW* dropping out by the intermediate stage (UFCO), and only *tetX* being present in the finished compost (FICO) and initial worm castings (TWCA). These particular Tetracycline resistance genes have been commonly found in other compost and manure samples, including swine [9] and cattle [37]. Only one ARG, Aminoglycoside resistance gene *aph(6)-1d* is present across all stages of composting until it is no longer detected in SWCA and WOCA samples. This gene is known to reside on plasmids and integrative elements and is capable of expression in both gram-positive and gram-negative species [38] indicating transfer across a variety of bacterial species and perhaps explaining its persistence throughout the composting cycle. As

**Table 2. Selected ARGs, known functions and associated samples.**

| Sample ID | Drug Class | Resistance Gene | Function |
|---|---|---|---|
| EGSH | Aminoglycoside | *aph(6) 1d* | Encodes streptomycin resistance via phosphotransferase enzyme |
| EGWA | | | Carried by plasmids, integrative conjugative elements, and chromosomal genomic islands in a variety of bacterial species [67] |
| FICO | | | Previously found in wastewater [33], |
| GROC | | | Present in both gram-positive and gram-negative species [68] |
| RFSC | | | |
| RWCO | | | |
| SCHO | | | |
| TWCA | | | |
| UFCO | | | |
| GROC | Macrolide | *lmrD* | Efflux pump utilizing ABC transporter [67, 69] |
| NURH | | | Chromosomally-encoded efflux pump; confers resistance to lincosamides |
| SCHO | | | Found primarily in *L. lactis* and *S. linconensis* |
| EGSH | Macrolide | *mefA* | Motive efflux pump conferring macrolide resistance [67] |
| EGWA | | | Found on an operon with *mefE* and *mel* |
| SCHO | | | Found in *S. pneumoniae* |
| EGSH | Macrolide | *mel* | A homolog of *msrA*, acts as an ABC transporter with macrolide resistance |
| EGWA | | | Expressed as an operon with *mefA* and *mefE* |
| SCHO | | | Found in *S. pneumoniae* |
| NURH | MDR Efflux pump | *abeM* | MATE pump family, extrudes aminoglycosides, fluoroquinolones, chloramphenicol, and more [67, 70] |
| | | | Found mainly in *A. baumannii* |
| NURH | MDR Efflux pump | *abeS* | Chromosomally-encoded efflux pump of SMR family, confers low-level resistance to multiple drugs & dyes [67, 71] |
| | | | Found mainly in *A. baumannii*, but present in *K. pneumoniae* |
| NURH | MDR Efflux pump | *adeF* | Complex of adeFGH operon; acts as RND efflux pump [67, 72] |
| | | *adeG* | Confers resistance to fluoroquinolone, tetracyline, tigecycline, chloramphenicol, clindamycin, trimethoprim, and sulfamethoxazole |
| | | *adeH* | Found mainly in *A. baumannii* |
| NURH | MDR Efflux pump | *adeI* | Complex of adeIJK operon; RND efflux pump [67, 73] |
| | | *adeJ* | Resistance to beta-lactams, chloramphenicol, tetracycline, erythromycin, lincosamides, fluoroquinolone, and more |
| | | *adeK* | Found mainly in *A. baumannii* |
| NURH | MDR Efflux pump | *emrD* | Efflux pump transporter from the MFS;; mainly found in E. coli [67, 74] |
| EGWA | Sulphonamide | *sul2* | Confers sulfonamide resistance via target replacement [67, 75, 76] |
| FICO | | | Present in wide range of gram-negative bacteria |
| RFSC | | | Notably present in *A. baumannii*, *K. pneumoniae*, and *S. enterica* |
| RWCO | | | |
| TWCA | | | |
| UFCO | | | |
| EGWA | Tetracycline | *tetH* | Tetracycline MFS efflux pump [31, 67] |
| FICO | | | Commonly linked to *sul2* and *strAB* |
| RFSC | | | Expressed in many gram-negative species, including *A. baumannii* |
| UFCO | | | Plasmid encoded, associated with tetR on pAST2 plasmid |
| EGSH | Tetracycline | *tetM* | Ribosomal protection protein conferring Tetracycline resistance; found on transposable elements [67, 77] |
| EGWA | | *tetO* | Found on conjugative plasmids [35] |

*(Continued)*

**Table 2.** (Continued)

| Sample ID | Drug Class | Resistance Gene | Function |
|---|---|---|---|
| RFSC | | | Associated with erythromycin resitance gene *ermB* |
| EGSH | Tetracycline | *tetW* | Ribosomal protection protein conferring Tetracycline resistance; present in both conjugative and non-conjugative elements |
| EGWA | | | Present in genera associated with the gut [78] |
| RFSC | | | Has been found in C. difficile [67] |
| UFCO | | | |
| EGWA | Tetracycline | *tetX* | Resistance to all clinically relevant tetracycline via an oxidoreductase activity that inactivates the drug [67, 79, 80] |
| RFSC | | | Found in anaerobic bacteria, particularly members of the genus *Bacteroides* |
| TWCA | | | |
| UFCO | | | |

Genes selected were present in multiple samples or conferred multidrug resistance (MDR).

such, it may make an ideal candidate for use as a marker gene of plasmid transfer in future studies.

## Virulence factors: Integrases, transposons, and enabling gene transfer

Fifty-four unique virulence factor associated genes were identified, with at least one being present in every sample type. The most frequently found were the genes *intl1*, *sul1*, and *tnpA*. Individual abundance ratios varied from $2.02^{-6}$ to 0.0402 and sample averages from 0.0002 to 0.056. While less abundant than ARGs identified, the total number of genes per sample was higher; an average of 9 virulence factor genes were found per sample compared to 7 ARGs.

Visualization of abundance ratio by heatmap displayed a more diffuse pattern of virulence gene abundance compared to ARGs (Fig 4). Low abundance carriage of multiple genes was common, especially among EGSH, EGWA, RFSC, UFCO, and UFCO. Of the virulence factors

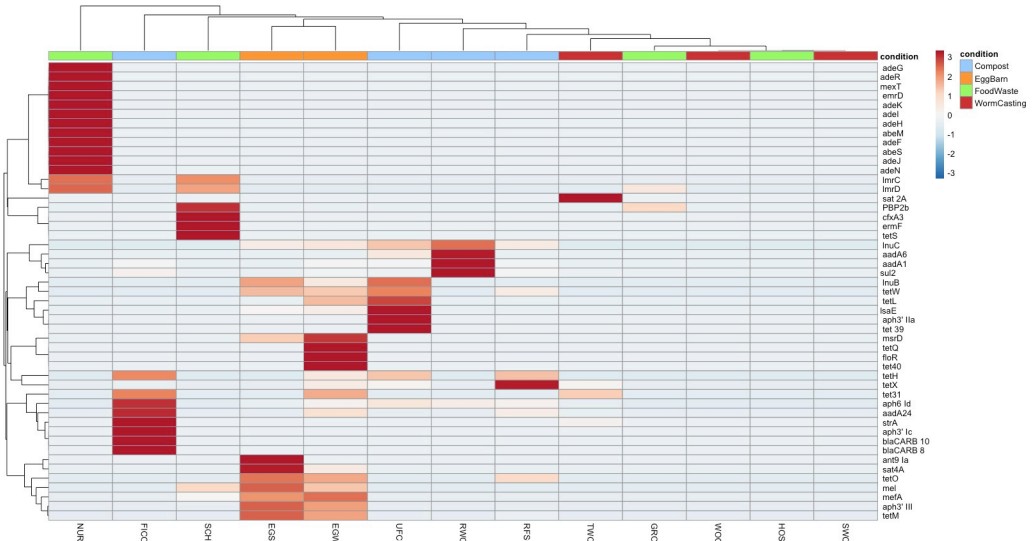

**Fig 3. Heatmap displaying the differences in abundance ratio of ARGs between samples.** Heatmap was scaled by row (individual ARGs) and created using the pheatmap package.

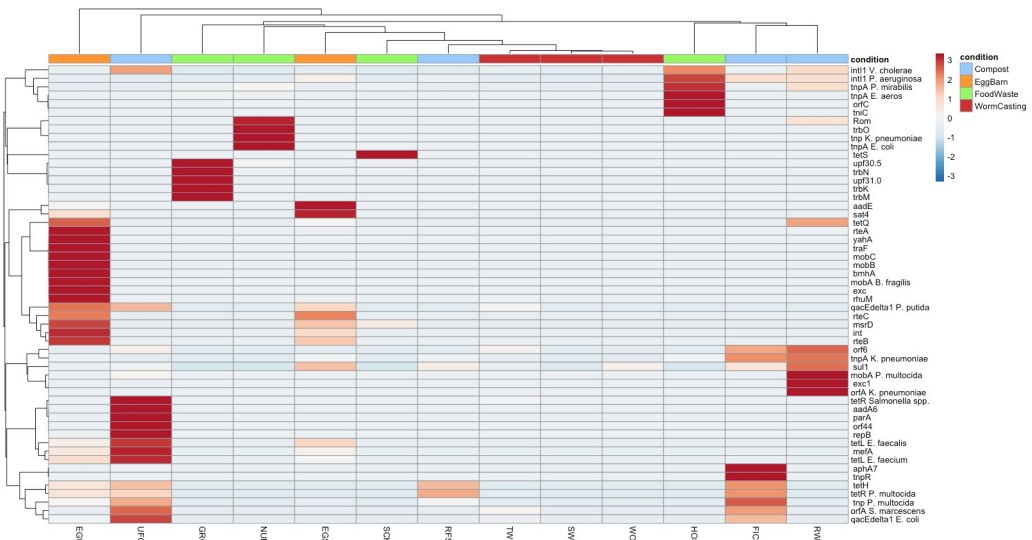

**Fig 4. Heatmap displaying the differences in abundance ratio of virulence factors between samples.** Heatmap was scaled by row (individual virulence genes) and created using the pheatmap package.

detected, several key integrases and a transposon regulator were identified (Table 3). *Intl1*, *tnpA* and *sul1* are commonly associated with the transfer of antimicrobial resistance [39,40] and may contribute to the transfer of ARGs within the farm setting, regardless of the survival or viability of the microorganisms that receive them.

## Microbial communities, niches, and EKSAPE pathogens

Microbial composition to the level of species or strain was accomplished using the CosmosID platform, a significant advantage over amplicon techniques. This allowed for not only the assessment of community structures and diversity, but also tracking of specific bacterial pathogens of concern.

Microbiome composition appears to be clustered both by sample type and composting stage (Fig 5). At the sample level, composition is driven most strongly by location (on-farm or a site-specific food waste) (Fig 5B), but variation can be seen between the various stages of composting as well (Fig 5A). Within the farm, distinct similarity can be seen between samples near the barn or in close contact with poultry (RFSC, EGSH, EGWA) and those at various stages of composting or vermicomposting. Additionally, worm casting samples after interaction with the worms (SWCA and WOCA) are very distinct from other composting samples. This is driven by the introduction of specific phyla seen only in these samples, including Thaumarchaeota, Verrucomicrobia, and Gemmatimonadetes. These have been prevalent in other vermicomposting studies [41–43]. In particular, Verrucomicrobia was found to correlate with cured composts [41] and be promoted by earthworms [42]. Other vermicomposting studies have indicated that dominant phyla may act as antagonists and help reduce pathogenic species [44].

The composition in relative abundance for the twenty most abundant species is shown in Fig 6A. The drivers behind the distances shown in Fig 6 can be traced to specific phyla here. As mentioned, vermicomposting samples contained several distinct phyla, but few of these were of high abundance overall. Notable species include *E. cecorum* in egg samples, *L. lactis* in food wastes, and the shifts in abundance of Actinobacteria species. Detection of *E. cecorum*, a

**Table 3. Selected virulence genes, function, associated organism and sample.**

| Sample ID | Associated Organism | Virulence Gene | Function |
|---|---|---|---|
| EGSH | *P. aeruginosa* | *intI1* | Integrase & resistance gene marker [81] |
| EGWA | | | Widely implicated in the spread of AMR; |
| FICO | | | has been detected in environmental phages, including |
| HOSP | | | soils and farms [82] |
| RFSC | | | |
| RWCO | | | |
| TWCA | | | |
| UFCO | | | |
| EGSH | *V. cholerae* | *intI1* | |
| EGWA | | | |
| FICO | | | |
| HOSP | | | |
| RFSC | | | |
| RWCO | | | |
| SWCA | | | |
| TWCA | | | |
| UFCO | | | |
| WOCA | | | |
| EGWA | *K. pneumoniae* | *orf6* | Contains domains related to cellular activities, |
| FICO | | | such as membrance fusion, proteolysis, and DNA |
| RWCO | | | replication [83] |
| SWCA | | | |
| TWCA | | | |
| UFCO | | | |
| WOCA | | | |
| EGSH | *P. mirabilis* | *sul1* | Linked to AMR genes carried on the same integron [84] |
| EGWA | | | Associated with presence of *aadA1* [59] |
| FICO | | | Associated with resistance genes in poultry meats [85] |
| RFSC | | | |
| RWCO | | | |
| SWCA | | | |
| TWCA | | | |
| UFCO | | | |
| WOCA | | | |
| EGWA | *P. multocida* | *tetR* | Transcriptional regulator of AMR [86] |
| FICO | | | |
| RFSC | | | |
| TWCA | | | |
| UFCO | | | |
| EGSH | *P. multocida* | *tnp* | Encodes transposase activity |
| EGWA | | | |
| FICO | | | |
| RFSC | | | |
| UFCO | | | |
| EGSH | *K. pneumoniae* | *tnpA* | Transposase gene |
| EGWA | | | Linked to strains carrying multiple ARGs [87] |
| FICO | | | |

*(Continued)*

**Table 3.** (Continued)

| Sample ID | Associated Organism | Virulence Gene | Function |
|---|---|---|---|
| HOSP | | | |
| RFSC | | | |
| RWCO | | | |
| TWCA | | | |
| UFCO | | | |
| EGWA | *P. mirabilis* | *tnpA* | |
| FICO | | | |
| GROC | | | |
| HOSP | | | |
| NURH | | | |
| RWCO | | | |
| SCHO | | | |

Genes selected were present in multiple samples.

known poultry commensal [45], and *L. lactis*, an additive during the fermentation of dairy products and other foods [46], in such specific niches highlights the sensitivity of shotgun sequencing for detection of bacterial species in a wide variety of food products. Additionally, by tracking the relative abundance of various members of the Actinobacteria phylum throughout composting a distinct compost profile emerges (Fig 6B). Species that were of low or zero abundance in food scraps or raw composts incrementally rise in abundance as composts matures or is transferred to the worm casting process, including *S. viridis*, *T. fusca*, and *M. thermoresistible*.

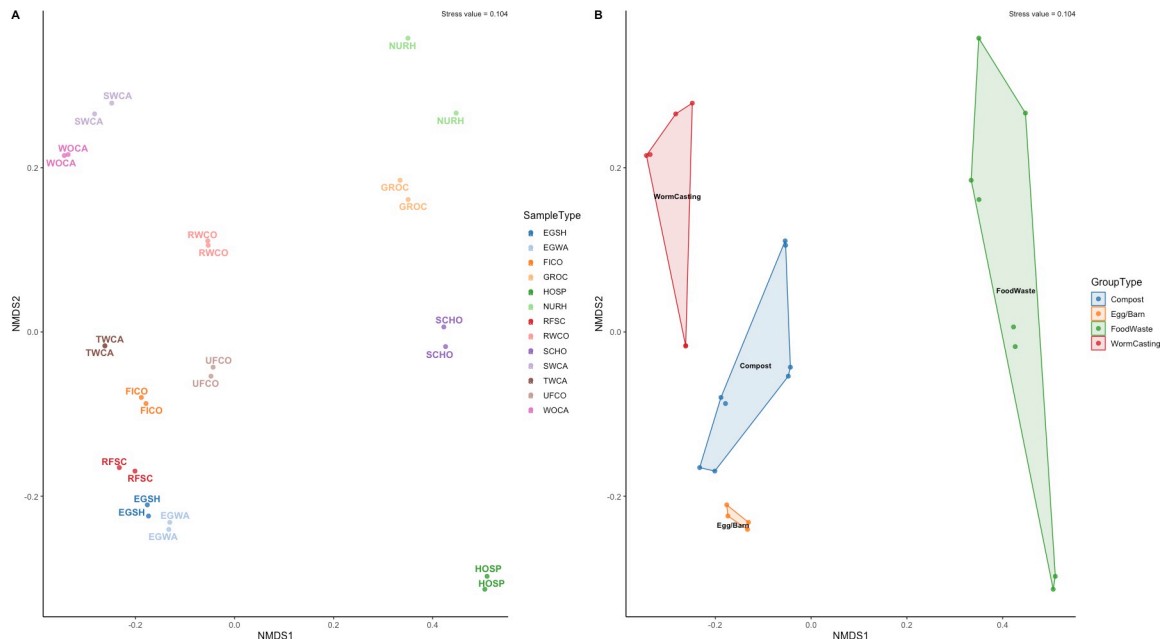

**Fig 5. Nonmetric Multidimensional Scaling (NMDS) plot using Bray-Curtis distance of the microbiome of each a) sample and b) group.** Food wastes refer to all samples collected off-farm.

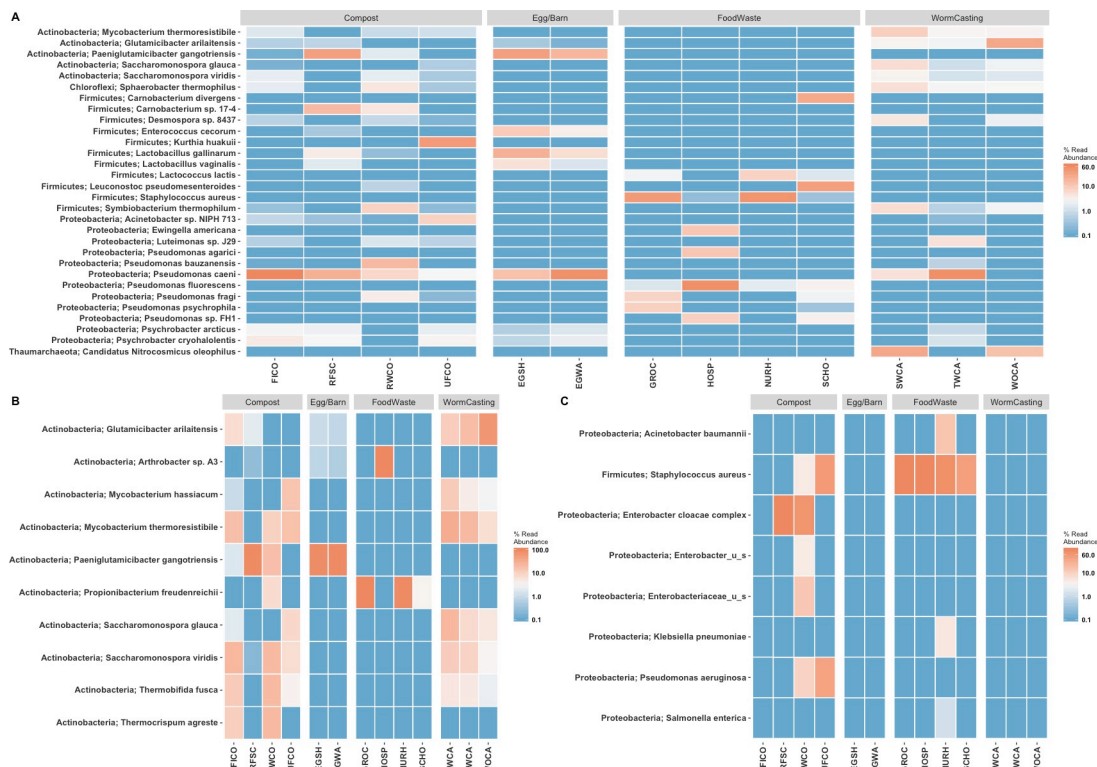

**Fig 6. Heatmap of the top A) 20 bacterial species in all samples, B) members of the Actinobacteria Phylum, and C) EKSCAPE pathogens**. The Phylum of each species precedes each species name. Heatmap is scaled as the log10 percent read abundance within each sample, with abundances <0.1% all being represented as the same color.

In addition to shifts in phyla abundance, specific strains and species can be tracked across samples due to the use of shotgun metagenomic sequencing. In terms of clinical infection risk, many surveillance efforts track the occurrence of ESKAPE pathogens. EKSAPE pathogens (*Enterococcus faecium*, *Staphylococcus aureus*, *Klebsiella pneumoniae*, *Acinetobacter baumanii*, *Pseudomonas aeruginosa*, and *Enterobacter* species) are responsible for the majority of nosocomial infections globally and can readily acquire antimicrobial resistance [47, 48]. Pathogens on this list were identified in several samples in this study but did not persist or occur in any samples that would be leaving the farm or used in agricultural land application (Fig 6C). *Klebsiella pneumoniae* and *Acinetobacter baumanii* were both isolated from the nursing home samples but were not present in any other materials.

*Salmonella enterica* was also present in food wastes from the nursing home, a species commonly causing severe food borne illness. Shell eggs are a potential source of food borne *Salmonella enterica* serovars Enteritidis and Typhimurium in humans, and salmonella control is an issue of particular concern for poultry farms. Salmonella control programs in laying flocks may include vaccination, introduction of new birds from salmonella free sources, environmental cleaning and disinfection procedures including rodent and fly control, treatment or decontamination of poultry feed, and a monitoring program for salmonella in the environment and eggs [49,50]. The farm in this study had previously completed limited monitoring of eggs for Salmonella using culture-based methods, identifying no egg contamination. Our results suggest metagenomic approaches might be used for pathogen monitoring in animal feeds, environmental samples, and products from farms [51,52]. As a result of our study findings the farmer has discontinued feeding of food wastes from the nursing home to the hens, and is exploring the potential of feeding food scraps

following an initial composting step. Further research is warranted to understand the application (including the diagnostic sensitivity and specificity) of next-generation sequencing approaches in on-farm food-borne pathogen monitoring and control.

*Staphylococcus aureus* was present in all four off-farm collection sites and the raw and unfinished composts. While not identified in any off-farm collection sites sampled at this single point in time, *Pseudomonas aeruginosa* and members of the *Enterobacteriaceae* family were identified in the raw and unfinished composts and raw food scraps and raw composts respectively. However, none of these appeared in the egg samples or finished compost products, indicating they are not a pressing risk to animal or environmental health. Only *S. aureus* was able to be characterized at the strain level, with strain MV8 being present in the majority of samples (sites and raw compost, excluding the unfinished compost). This strain has been identified as sequence type (ST) 8 and containing a derivative of the SCC*mec* IV element responsible for methicillin resistance [53]. Other isolates of this group (ST 8) have been identified globally in cases of community-acquired methicillin-resistant *S. aureus* infections (CA-MRSA), such as USA 300 throughout the United States and CA-MRSA/J in Japan [54]. The disappearance or removal below detectable levels of this strain is promising evidence for the attenuation of EKSAPE pathogens by the composting process.

## Association between resistome and virulome

Transfer of specific genes or species appears to be rare between collection sites and farm samples. Only 3 ARGs, 9 virulence factors, and 18 bacterial species were found in both a collection site and any on-farm material, which may indicate successful mitigation by the composting process as seen in other studies [55]. However, only four collection sites were sampled, so additional analyses were performed to assess the relationship between bacterial composition and persistence of antibiotic resistance genes.

Prior work has demonstrated a relationship between antibiotic resistance genes and associated sample microbiome [56,57]. To investigate this potential relationship, Pearson correlations between richness, Shannon and Simpson diversities, and ARG counts and diversity were performed (Tables 4 and 5). None of these proved significant however, which prompted the investigation of potential interactions between resistance genes and virulence genes facilitating

**Table 4. Summary of diversity metrics for each sample.**

| | Bacteria | | | ARG | | | VF | |
|---|---|---|---|---|---|---|---|---|
| Sample | Richness | Shannon Diversity | Simpson Diversity | Count | Shannon Diversity | Simpson Diversity | Shannon Diversity | Simpson Diversity |
| EGSH | 37.50 | 2.10 | 0.78 | 12 | 1.84 | 0.80 | 2.24 | 0.88 |
| EGWA | 60.00 | 2.58 | 0.83 | 21 | 2.42 | 0.89 | 2.66 | 0.91 |
| FICO | 54.00 | 3.34 | 0.95 | 10 | 0.95 | 0.49 | 1.79 | 0.80 |
| RFSC | 59.00 | 2.31 | 0.78 | 9 | 1.80 | 0.81 | 1.30 | 0.66 |
| RWCO | 65.50 | 3.41 | 0.94 | 4 | 0.77 | 0.39 | 1.07 | 0.54 |
| UFCO | 129.50 | 2.87 | 0.81 | 13 | 1.85 | 0.80 | 2.33 | 0.88 |
| TWCA | 43.00 | 3.42 | 0.96 | 6 | 1.23 | 0.68 | 1.80 | 0.76 |
| SWCA | 17.50 | 2.33 | 0.85 | 0 | 0.00 | 1.00 | 0.74 | 0.45 |
| WOCA | 36.50 | 2.84 | 0.88 | 0 | 0.00 | 1.00 | 0.61 | 0.42 |
| GROC | 42.00 | 2.44 | 0.81 | 3 | 0.93 | 0.57 | 1.47 | 0.71 |
| HOSP | 54.50 | 1.74 | 0.66 | 0 | 0.00 | 1.00 | 1.14 | 0.56 |
| NURH | 19.50 | 1.37 | 0.50 | 15 | 2.39 | 0.89 | 0.93 | 0.46 |
| SCHO | 85.00 | 2.48 | 0.81 | 8 | 1.24 | 0.58 | 0.23 | 0.11 |

Measurements were taken across replicates and averaged. Richness, Shannon, and Simpson diversity were all calculated using the vegan package in R.

**Table 5. Results of Pearson correlation testing.**

| Relationship | Correlation Coefficient | p-value |
|---|---|---|
| Bacteria Richness * ARG Count | 0.273 | 0.37 |
| Bacteria Shannon Diversity * ARG Count | -0.112 | 0.72 |
| Bacteria Simpson Diversity *ARG Count | -0.235 | 0.39 |
| Bacteria Shannon Diversity * ARG Shannon Diversity | -0.19 | 0.53 |
| Bacteria Simpson Diversity * ARG Simpson Diversity | -0.486 | 0.09 |
| Bacteria Shannon Diversity * VF Shannon Diversity | 0.204 | 0.50 |
| Bacteria Simpson Diversity * VF Simpson Diversity | 0.203 | 0.51 |
| ARG Shannon Diversity * VF Shannon Diversity | 0.553 | 0.05** |
| ARG Simpson Diversity * VF Simpson Diversity | -0.038 | 0.91 |

All tests were conducted using the Hmisc package in R

** denotes statistical significance.

gene transfer events. Co-occurrence of virulence genes and antibiotic resistance has been shown in *Pseudomonas aeruginosa* [58] and has a stronger association than antibiotic use alone in populations of *E. coli* [59,60]. In the current study, this relationship between antibiotic resistance genes and virulence factors produced the only statistically significant result, with Shannon diversities of these gene categories being positively correlated (0.553, $p = 0.05$).

This relationship was further explored through co-inertia analysis. Briefly, co-inertia analysis is a multivariate method that can robustly couple tables, ecological or otherwise, given time points or samples are shared across measured variables [61]. For example, this technique has been applied to soil ecology studies, assessing patterns of syntony in samples across environmental characteristics such as pH or temperature with microbial communities or species. The main benefit of co-inertia analysis over similar techniques such as redundancy analysis (RDA) or canonical correspondence analysis (CCA), is that it is not constrained by the number of variables or observations. Thus, it is capable of measuring the global co-structure between two sets of variables regardless of if they can be measured on a gradient. In this study, it was applied to assess the similarity between patterns of microbial communities and functional genes (ARGs and virulence genes); results are expressed on a scale of 0 to 1, which 0 being unrelated and 1 being strong patterns of covariance. The results of co-inertia analyses provided further evidence of syntony between resistance and virulence genes (RV = 0.647), compared to 0.445 between that of bacterial communities and ARGs and 0.358 between bacteria and virulence genes. Similar mechanisms of regulation and induction, such as biofilm formation, communication, and HGT have been implicated in the link between resistance and virulence genes [62].

These results may shed light on the dynamics of ARG transfer specifically within the composting environment; large population shifts occurred during thermophilic phases, but the genes regulating gene transfer are more consistent. Notably, in samples where no ARGs were identified (WOCA, SWCA, HOSP) fewer virulence genes were present. Both SWCA and WOCA carried only *sul1*, *intl1*, and *orf6* and HOSP contained *intl1*, *orfC*, *tniC*, and *tnpA*. Conversely, samples with the most ARGs (EGWA, EGSH, NURH, and UFCO) contained 26, 15, 6, and 16 virulence genes respectively.

Alternatively, differentiation between total microbial community and so-called reservoir hosts should be explored. Wang et al., investigated this relationship using both metagenomic and metatranscriptomic data in a controlled setting to elucidate the effects of composting stage on resistome profile [63]. While core resistome profiles were quite stable, the core ARGs were carried by different bacterial phyla as environmental conditions changed during the composting stages; this succession of phyla maintaining core ARG families may be happening in

food waste composting and may be responsible for the relationships identified in our study. Identification of these reservoir hosts should be conducted in further sampling efforts in addition to characterization of important virulence or functional genes facilitating ARG persistence.

## Limitations of the current study

There are a number of limitations of our current study that may help explain some of the findings. First, due to financial and logistical constraints we completed sampling at a single point in time on a single farm across the farm system and food waste collection sites. A longitudinal study design repeatedly sampling all sites over time would improve our understanding of ARG dynamics in the system and provide improved evidence of differences associated with steps in the food waste processing. This may be especially important to confirm our finding of a reduction in ARGs during the final steps of composting and vermiculture. Similarly, extending this study to multiple farms would be a critical next step, and we are aware of at least 3 other diversified farms in Vermont that are feeding comingled post-consumer and institutional food waste to commercial poultry flocks. We also did not include direct sampling of the poultry or other animals on the farm and limited our samples to materials including eggs collected from the chicken house. Future studies might include fecal and skin sampling of the poultry (e.g. cloacal and skin swabs), however in this study we restricted samples to egg samples (external egg wash and shell) due to convenience, elimination of the requirement for animal institutional animal care and use committee approval, and relevance as a food product sold from this farm. A more comprehensive study would include culture-based analysis of the samples in order to identify, isolate and quantify resistant bacteria across the farm system. Screening samples for a panel of antimicrobials representing all classes can be labor intensive. We propose that among the strengths of the metagenomic approach is it can be used in series with culture-based methods; where the genomic results can be used to inform subsequent selective culture media choice for isolation of phenotypically resistant organisms from stored samples.

Alternatively, where a culturing approach may still be cost prohibitive or unsuccessful for environmental microbes, recent advances in long-read sequencing for metagenomics position this method as a viable option. The main advantage of long-read sequencing is the illumination of genetic context; by sequencing longer read fragments, users are not limited to simply detecting the presence of ARGs but can also identify gene clusters that show co-occurrence with other resistance genes or mobile elements that are the likely mechanisms of transfer. Specific tools have been developed for this purpose, such as the accessible web service NanoARG [64] or the interactive software MEGAN-LR [65]. These new sequencing technologies have already been used for rapid detection of ARGs, where the Oxford Nanopore MinION was successfully used in a veterinary hospital to detect ARGs, assess their taxonomy and origin, and to inform mitigation strategies [65,66]. Long-read sequencing should provide contextual information for ARGs found in multiple stages of the composting process. For example, while *aph(6) 1d* was found in multiple samples in this study, we cannot determine if a single organism is carrying this gene or if it is being transferred via a mobile element to the microbial community along the composting stream. The use of long-read sequencing will likely become common place in future studies of this kind, as the MinION device is well-suited to a field setting due to its portability needs and could be an valuable tool in guiding bioremediation throughout the food waste composting cycle.

## Conclusion

The aim of this work was to identify, characterize, and provide insight into the dynamics of antibiotic resistance genes during food waste management on a farm feeding food scraps to poultry and subsequently composting the material. Using shotgun metagenomic sequencing

we were able to accomplish this by evaluating the microbiome, resistome, and relevant functional genes of collected samples. While limited to a single farm, these results indicate that ARGs and pathogenic bacterial species are reduced in both number and abundance during the food waste composting process, recapitulating results shown in manure composting operations and expanding knowledge of this important management practice. Notably, the relationship between virulence factors and antibiotic resistance genes should be further explored and may be key in preventing additional spread of ARGs throughout the food waste composting process and at the commercial scale. Given the driver of improved agricultural sustainability, we expect the import of commercial food waste on to farm systems is likely to increase in the near future, including on small to medium sized farms such as the one in this study. Research should focus on expanding this work to additional farming systems and compost management styles to fully assess the associated risk; this work provides an accessible analytical framework and baseline data for future studies.

## Supporting information

**S1 Table. ARG sequences frequency and abundance identified by sample after sequence read filtering.**
(XLSX)

**S2 Table. Virulence factor sequences frequency and abundance identified by sample after sequence read filtering.**
(XLSX)

**S3 Table. Bacterial taxa sequences frequency and abundance identified by sample after sequence read filtering.**
(XLSX)

## Acknowledgments

The authors would like to acknowledge the Vermont Integrative Genomics Resource for technical support and sequencing efforts. We would like to thank the collaborating farmer for providing access for sample collection. We also extend thanks to Willa Richmond for additional laboratory work.

## Author Contributions

**Conceptualization:** John W. Barlow.

**Data curation:** John W. Barlow.

**Formal analysis:** Korin Eckstrom.

**Funding acquisition:** Korin Eckstrom, John W. Barlow.

**Investigation:** Korin Eckstrom, John W. Barlow.

**Methodology:** Korin Eckstrom, John W. Barlow.

**Project administration:** John W. Barlow.

**Resources:** John W. Barlow.

**Supervision:** John W. Barlow.

**Validation:** Korin Eckstrom.

**Visualization:** Korin Eckstrom.

**Writing – original draft:** Korin Eckstrom.

**Writing – review & editing:** John W. Barlow.

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
