## [Decision Letter · Decision Letter 0]

10 Oct 2019

PONE-D-19-18208

Resistome metagenomics from plate to farm: the resistome and microbial composition during food waste feeding and composting on a Vermont poultry farm

PLOS ONE

Dear Dr. Barlow,

Thank you for submitting your manuscript to PLOS ONE. After careful consideration, we feel that it has merit but does not fully meet PLOS ONE’s publication criteria as it currently stands. Therefore, we invite you to submit a revised version of the manuscript that addresses the points raised during the review process.

Your manuscript has been reviewed by three experts in your field.  A minor revision is suggested by one of the reviewers.

We would appreciate receiving your revised manuscript by 2 weeks. To enhance the reproducibility of your results, we recommend that if applicable you deposit your laboratory protocols in protocols.io, where a protocol can be assigned its own identifier (DOI) such that it can be cited independently in the future. For instructions see: http://journals.plos.org/plosone/s/submission-guidelines#loc-laboratory-protocols

We look forward to receiving your revised manuscript.

Kind regards,

Yung-Fu Chang

Academic Editor

PLOS ONE

Journal Requirements:

2. In your Methods section, please provide additional location information, including geographic coordinates for the data set if available.

Additional Editor Comments (if provided):

Reviewers' comments:

Reviewer's Responses to Questions

**Comments to the Author**

1. Is the manuscript technically sound, and do the data support the conclusions?

Reviewer #1: Yes

Reviewer #2: Yes

Reviewer #3: Yes

2. Has the statistical analysis been performed appropriately and rigorously? 

Reviewer #1: Yes

Reviewer #2: Yes

Reviewer #3: Yes

3. Have the authors made all data underlying the findings in their manuscript fully available?

Reviewer #1: Yes

Reviewer #2: Yes

Reviewer #3: Yes

4. Is the manuscript presented in an intelligible fashion and written in standard English?

Reviewer #1: Yes

Reviewer #2: Yes

Reviewer #3: Yes

5. Review Comments to the Author

Reviewer #1: Well written paper and asking key questions about ARG movement in food system.

Limitations of the study are addressed.

No technical issues with approach.

Conclusions are appropriate.

Reviewer #2: I think this is an excellent piece of research which is explained in great detail and written to a very high standard.

I would suggest discussing the use of host depletion methods rather than simply filtering human reads from your data. Also, rather than coupling metagenomics back with culture you could explain the advantages of using long-read sequencing. This would also give further insight into the spread of the AMR genes you have identified and give context as to whether the genes are shared among species of if the same species harbouring the AMR gene is found throughout the sampling/composting process.

With regards the figures, they need to be made to a higher quality as the text can not be read as they stand.

Reviewer #3: This manuscript describes the impact of feeding farmed animals with human food scraps on the dissemination of antimicrobial resistance within the farm products and compost materials. The authors used metagenomics to explore the bacterial resistome, microbiome and virulence markers across different sites of a layer chicken farm, as well as in the food scraps that were fed to the birds. The DNA sequence analysis relied on a commercial software suite (CosmosID) which provides an integrated pipeline for processing datasets. Conventional statistical approaches for descriptive community ecology were used.

This is an interesting investigation. The technical approaches used here may be applied to a number of small-to-medium scale operations across the world, given the growing interest in the development of sustainable practices for agriculture.

The paper is clearly written and the results are convincing, albeit restricted to one farm as a case study.

Major remarks:

The apparent presence of Salmonella in some of the food scraps given to the chicken is worrying, given the zoonotic potential of this organism. This may be of particular importance as egg products can transmit some serovars (Enteritidis, Typhimurium) to humans. Some farms have surveillance programs for the detection of environmental Salmonella sp.; could the author comment on this point?

The use of antimicrobial for treatments or prevention of infectious diseases in the farm is not described in the manuscript. Relatively few antibiotics are approved in commercial layers but the situation varies depending on local regulations. This may have an impact on the nature and abundance of ARGs detected on the flora present on the eggs. Could the author comment on this point?

Minor remarks:

L356 Actinbacteria should read Actinobacteria

L440-441 The sentence “(…) they were able to identify different bacterial of these ARGs across stages (…)” is unclear. Do you mean “different bacterial origins of these ARGs”?

6. PLOS authors have the option to publish the peer review history of their article (what does this mean?). If published, this will include your full peer review and any attached files.

Reviewer #1: Yes: James F Lowe

Reviewer #2: No

Reviewer #3: No

---

## [Author Response · Author response to Decision Letter 0]

25 Oct 2019

Journal Requirements:

AU: My apologies that we did not pay closer attention to style requirements in the initial submission. We have addressed that issue in this revision.

2. In your Methods section, please provide additional location information, including geographic coordinates for the data set if available.

AU: we have added additional location information to the methods section at line 104 in the full mark-up version.

AU: we have added additional research compliance information to the methods section at line 132 in the full mark-up version.

Reviewer #1: Well written paper and asking key questions about ARG movement in food system.

Limitations of the study are addressed.

No technical issues with approach.

Conclusions are appropriate.

AU: we thank this reviewer for their review of our manuscript.

Reviewer #2: I think this is an excellent piece of research which is explained in great detail and written to a very high standard.

I would suggest discussing the use of host depletion methods rather than simply filtering human reads from your data. 

AU: We agree this is an important concept to highlight, and have added 2 sentences at lines 141 – 146 to emphasize the physical host tissue depletion methods. We have also added sentences at lines 146-156 in the methods briefly describing our initial pilot experiments to optimize DNA yields while reducing host DNA contamination. In conjunction with these additions we have made minor changes in the subsequent methods text (line 157 and line 178) to provide clarification and eliminate repetition in the text. We have also added four sentences to the discussion section (line 275-283) highlighting our thoughts on physical host tissue depletion methods for complex samples and the limitations in our preliminary experiments using filtration.

Also, rather than coupling metagenomics back with culture you could explain the advantages of using long-read sequencing. This would also give further insight into the spread of the AMR genes you have identified and give context as to whether the genes are shared among species of if the same species harbouring the AMR gene is found throughout the sampling/composting process.

AU: Thank you for this comment. Since completing this project, author KE has more recent experience with long-read sequencing and we agree this is an excellent alternative to linking resistance genes to species markers for our future studies. We have added a paragraph discussing this concept, including 2 additional references, at lines 541 to 558.

With regards the figures, they need to be made to a higher quality as the text can not be read as they stand.

AU: Thank you, we agree and have increased the font size of figures

Reviewer #3: This manuscript describes the impact of feeding farmed animals with human food scraps on the dissemination of antimicrobial resistance within the farm products and compost materials. The authors used metagenomics to explore the bacterial resistome, microbiome and virulence markers across different sites of a layer chicken farm, as well as in the food scraps that were fed to the birds. The DNA sequence analysis relied on a commercial software suite (CosmosID) which provides an integrated pipeline for processing datasets. Conventional statistical approaches for descriptive community ecology were used.

This is an interesting investigation. The technical approaches used here may be applied to a number of small-to-medium scale operations across the world, given the growing interest in the development of sustainable practices for agriculture.

The paper is clearly written and the results are convincing, albeit restricted to one farm as a case study.

Major remarks:

AU: We thank reviewer 3 for bringing forward the 2 critical issues of public health implications and antimicrobial use on the study farm. We agree these issues should be addressed in the discussion. We also appreciate their recognition of the application of the methods to small-to-medium scale agricultural operations in the context of growing need for developing sustainable agricultural practices. We have incorporate similar language at line 571 in the conclusion.

The apparent presence of Salmonella in some of the food scraps given to the chicken is worrying, given the zoonotic potential of this organism. This may be of particular importance as egg products can transmit some serovars (Enteritidis, Typhimurium) to humans. Some farms have surveillance programs for the detection of environmental Salmonella sp.; could the author comment on this point?

AU: We and the farmer agree about the worrying aspect of this finding. We have added a paragraph addressing these concerns, and the potential application of metagenomic approaches in surveillance programs (including reference to 2 recent review articles) at lines 427-440. We conclude this section with a note (anecdote) that the farmer subsequently used our results to make management changes to their system by discontinuing feeding the chickens food scraps from the nursing home, and diverting this waste directly to the composting stream.

The use of antimicrobial for treatments or prevention of infectious diseases in the farm is not described in the manuscript. Relatively few antibiotics are approved in commercial layers but the situation varies depending on local regulations. This may have an impact on the nature and abundance of ARGs detected on the flora present on the eggs. Could the author comment on this point?

AU: We added a sentence at line 107 in the methods section “With the exception of feeding post-consumer food waste, which can not be certified as an organic feed, the farm follows USDA National Organic Program standards for animal husbandry practices, including no use of antimicrobials.” And in the discussion at line 328 we have added more detail on a prior study, and at line 330 we have added two sentences discussing our findings relative to no-antimicrobial use on the study herd, and the limitation of our sampling scope in space and time.

Minor remarks:

L356 Actinbacteria should read Actinobacteria

AU: corrected

L440-441 The sentence “(…) they were able to identify different bacterial of these ARGs across stages (…)” is unclear. Do you mean “different bacterial origins of these ARGs”?

AU: we have revised this sentence now at line 509, “While core resistome profiles were quite stable in composition, the core ARGs were carried by different bacterial phyla as environmental conditions changed during the composting stages; this succession of phyla maintaining core ARG families may be happening in food waste composting as well and may be responsible for the relationships identified in our study.”

---

## [Editor Report · Decision Letter 1]

1 Nov 2019

Resistome metagenomics from plate to farm: the resistome and microbial composition during food waste feeding and composting on a Vermont poultry farm

PONE-D-19-18208R1

Dear Dr. Barlow,

We are pleased to inform you that your manuscript has been judged scientifically suitable for publication and will be formally accepted for publication once it complies with all outstanding technical requirements.

With kind regards,

Yung-Fu Chang

Academic Editor

PLOS ONE
---

## [Editor Report · Acceptance letter]

12 Nov 2019

PONE-D-19-18208R1 

Resistome metagenomics from plate to farm: the resistome and microbial composition during food waste feeding and composting on a Vermont poultry farm 

Dear Dr. Barlow:

I am pleased to inform you that your manuscript has been deemed suitable for publication in PLOS ONE. Congratulations! Your manuscript is now with our production department. 

With kind regards,

on behalf of

Dr. Yung-Fu Chang 

Academic Editor

PLOS ONE